# Effectiveness and Safety of CGRP-Targeted Therapies Combined with Lifestyle Modifications for Chronic Migraine in Korean Pediatric Patients: A Retrospective Study

**DOI:** 10.3390/brainsci15050493

**Published:** 2025-05-08

**Authors:** Ji-Hoon Na, Hayoon Jeon, Ji-Eun Shim, Hyunjoo Lee, Young-Mock Lee

**Affiliations:** Department of Pediatrics, Gangnam Severance Hospital, Yonsei University College of Medicine, Seoul 135-720, Republic of Korea; jhnamd83@yuhs.ac (J.-H.N.);

**Keywords:** pediatric migraine, chronic migraine, calcitonin gene-related peptide (CGRP), CGRP monoclonal antibody, lifestyle modification

## Abstract

**Background/Objectives:** Pediatric chronic migraine (CM) is a debilitating condition with challenging management due to diagnostic complexities and a lack of evidence-based treatment. Calcitonin gene-related peptide (CGRP)-targeted therapies have transformed adult CM management, but their use in pediatric populations is underexplored. This study evaluated the safety and efficacy of CGRP-targeted therapies combined with structured lifestyle modifications in Korean pediatric patients with CM. **Methods:** This retrospective study examined 10 pediatric CM patients treated at Gangnam Severance Hospital from 2021 to 2024. Inclusion criteria were as follows: (1) Pediatric Migraine Disability Assessment Scale (PedMIDAS) score ≥ 30, (2) >2 failed preventive therapies, and (3) ≥8 migraine days per month. Patients received CGRP monoclonal antibodies or antagonists, alongside sleep, dietary, and exercise interventions. Changes in migraine burden, neuropsychological outcomes, and adherence to lifestyle interventions were assessed over 12 months. **Results:** Migraine frequency significantly decreased from a median of 26.5 to 14 days per month (*p* < 0.001); PedMIDAS scores declined from 58.5 to 48.0 (*p* = 0.037); and acute analgesic use was reduced from 14 to 5 days per month (*p* < 0.001). Adherence to lifestyle interventions improved significantly (*p* < 0.001). No serious adverse events were reported, and minor side effects, such as injection site pain and dizziness, were self-limiting. **Conclusions:** CGRP-targeted therapies, combined with structured lifestyle modifications, safely and effectively reduce migraine burden in pediatric CM patients. These therapies have facilitated sustainable improvements in management and support their integration into comprehensive pediatric CM care. This study highlights the importance of integrating pharmacologic and lifestyle-based approaches for holistic pediatric migraine management.

## 1. Introduction

Pediatric chronic migraine (CM) is a prevalent yet often underdiagnosed neurological disorder, imposing substantial burden on affected children and adolescents worldwide [1,2]. Pediatric CM is characterized by the occurrence of headaches more than 15 days per month for at least 3 months and can severely disrupt academic performance, social interactions, and emotional well-being [1,3]. It is underrecognized, as pediatric patients often present with atypical features and have difficulty articulating headache characteristics. Despite its prevalence, accurate diagnosis remains elusive due to a lack of specific biomarkers and the reliance on subjective clinical criteria. Moreover, treatment options are often extrapolated from adult CM or general headache therapies, which are not migraine-specific and lack robust evidence for efficacy in pediatric populations [4,5]. These factors contribute to the global trend of overlooking and underdiagnosing pediatric CM, resulting in inadequate treatment in many patients.

Recent advancements in migraine research have highlighted the central role of calcitonin gene-related peptide (CGRP) in the pathophysiology of CM [6]. CGRP-targeted therapies, including CGRP monoclonal antibodies (CGRP mAbs) and antagonists, have revolutionized the prophylactic treatment of adult CM by significantly reducing headache frequency and improving patient functionality. Their use marks a shift toward mechanism-based therapies that target the specific pathways involved in migraines [7,8]. While CGRP-targeted therapies are widely used in adult populations, their application in pediatric patients remains largely experimental. Given limited treatment options for pediatric CM, CGRP-targeted therapies are emerging as promising alternatives [9]. Since 2018, pediatric headache specialists have advocated for their use in carefully selected cases, with emerging studies showing promising outcomes regarding safety, efficacy, and functional improvements [10,11,12,13,14].

To our knowledge, this is the first Korean study to investigate the safety and efficacy of CGRP-targeted therapies in pediatric patients with CM. In Korea, the prevalence of migraine among adolescents is estimated at approximately 10%, yet limited data exist regarding chronic migraine in this population. Further, we explored the utility of CGRP-targeted therapies as transitional therapies to facilitate meaningful lifestyle modifications [1,15]. By providing preliminary but clinically informative evidence of their clinical utility, this research aims to advance the understanding of pediatric CM treatment using CGRP-targeted therapies and to establish a foundation for integrating innovative biologic therapies into routine pediatric neurological practice [1,16].

## 2. Materials and Methods

### 2.1. Study Design and Participants

This study is a retrospective analysis of prospectively collected clinical data, conducted at the Pediatric Headache Center of Gangnam Severance Hospital, Yonsei University College of Medicine, and includes pediatric patients diagnosed with migraine between March 2021 and September 2024. The eligibility criteria included experiencing at least 8 headache days per month, a Pediatric Migraine Disability Assessment Scale (PedMIDAS) score of ≥30, and failure of at least two preventive therapies. Patients aged ≥12 years were included if they had no contraindications, such as significant cardiovascular or neurological comorbidities [10,12,14].

During this period, 173 pediatric patients were identified as having migraines. Of these, 131 were diagnosed with episodic migraines and were excluded from this study. The remaining 42 patients met the criteria for CM as defined by the International Classification of Headache Disorders, Third Edition (ICHD-3). This includes headache occurrence on 15 or more days per month for at least 3 months, with at least 8 of these days meeting the criteria for migraine [17].

Of the 42 patients with CM, 18 met the additional inclusion criteria, which included a PedMIDAS score of 30 or higher and failure of at least two preventive therapies [18]. These 18 patients received treatment with either CGRP mAbs or antagonists, combined with stepwise lifestyle modifications. Of the 18 eligible patients, 8 were excluded due to insufficient follow-up data, resulting in 10 patients included in the final analysis. To ensure accuracy and consistency, all diagnoses were confirmed by two pediatric neurologists specializing in headache disorders. This selection process was designed to focus on evaluating the effectiveness of CGRP-targeted therapies combined with lifestyle modifications in pediatric patients with CM who were treated at a specialized headache center (Figure 1).

### 2.2. Ethics Approval and Patient Consent

This study was conducted in strict accordance with the ethical principles outlined in the Declaration of Helsinki. Institutional Review Board (IRB) approval was obtained from Gangnam Severance Hospital, Yonsei University College of Medicine (approval number: 3-2023-0254). Given the retrospective nature of this study, the IRB waived the requirement for written informed consent. All data collection and analyses were conducted in compliance with local legislation and institutional requirements to ensure patient privacy and data confidentiality. Patient data were de-identified prior to analysis to safeguard personal information.

### 2.3. Data Collection

Data were collected through a comprehensive review of medical records, focusing on a wide range of variables that characterize patients’ migraines. Baseline data were collected from structured interviews and full clinical history obtained at initial consultation. Baseline demographic and clinical characteristics, including patient age, sex, body weight, headache duration (refers to the time elapsed since the initial onset of headache, measured in months or years), and sleep patterns, were documented. Detailed headache characteristics, such as headache frequency (number of headache days per month), headache intensity (rated on a 0–10 Visual Analog Scale (VAS)), typical duration of episodes, and associated symptoms, were recorded. Headache episode duration was measured in hours. Analgesic use was quantified as the average number of days per month on which pain-relief medications were used. Additionally, data on the type of CGRP mAbs or prescribed agents were collected, including specific medications such as galcanezumab, fremanezumab, and atogepant. Galcanezumab, fremanezumab, and atogepant were chosen based on clinical availability and safety data in adolescent populations; erenumab and eptinezumab were not available or approved in South Korea at the time. Information on treatment-related adverse effects, as reported by the patients or documented in medical records, was systematically gathered to evaluate tolerability and safety [13,14].

Follow-up data were obtained during scheduled prospective outpatient visits every 3 months, during which changes in migraine intensity and frequency were monitored using patient-maintained headache diaries and clinician documentation, based on structured questionnaires and clinical assessments. This approach ensured that the data were accurate and reflective of the patients’ experiences throughout the treatment course. By capturing these comprehensive variables, this study provided a detailed overview of patients’ migraine profiles and their responses to CGRP-targeted therapies.

### 2.4. Headache Questionnaires and Neuropsychological Tests

To comprehensively evaluate the impact of migraines on daily life and overall neuropsychological status, a variety of validated assessment tools were employed. Headache-related disability was assessed using the Pediatric Migraine Disability Assessment Scale (PedMIDAS). According to established severity cutoffs, PedMIDAS scores are interpreted as follows: mild disability (<11), moderate (11–30), severe (31–50), and very severe (>50). To evaluate emotional and psychological well-being, the Children’s Depression Inventory (CDI) provided a detailed analysis of depressive symptoms, the Revised Children’s Manifest Anxiety Scale (RCMAS) measured anxiety symptoms, and the Generalized Anxiety Disorder-7 (GAD-7) scale assessed generalized anxiety. The Patient Health Questionnaire-9 (PHQ-9) was also used to screen for depressive symptoms. The combination of screening tools (PHQ-9, GAD-7) and pediatric-specific instruments (CDI, RCMAS) allowed comprehensive assessment across age-appropriate domains. Assessments were performed at baseline and at 3-month intervals to monitor longitudinal changes in headache-related disability, cognitive function, and emotional well-being [19,20].

Comprehensive neuropsychological testing was conducted to evaluate cognition, memory, and social functioning. This included the Wechsler Intelligence Scale for Children, Fifth Edition (WISC-V), which assesses the Full-scale Intelligence Quotient (FSIQ), Memory Quotient (MQ) to evaluate memory function, and Social Quotient (SQ) to measure social development and adaptability. These tests provide a holistic view of the patients’ neuropsychological status.

### 2.5. Neuroimaging, Electroencephalography, and Genetic Testing

All patients underwent brain magnetic resonance imaging (MRI) and magnetic resonance angiography (MRA) as part of the diagnostic workup to exclude secondary causes of headache, such as structural abnormalities or vascular pathology, in accordance with institutional protocol given the chronicity and refractoriness of their symptoms [17]. These imaging studies were interpreted by neuroradiologists with expertise in pediatric imaging to ensure the identification of underlying conditions that might contraindicate treatment with CGRP-targeted therapies.

Additional diagnostic evaluations were performed to rule out epilepsy and genetic migraine syndromes in patients with migraines and auras (n = 4). Electroencephalography (EEG) was conducted to detect potential epileptiform activity, and genetic tests were performed to identify mutations associated with hereditary migraine conditions, such as familial hemiplegic migraine. These tests are particularly important for patients with atypical or complex presentations to ensure an accurate diagnosis and appropriate treatment selection.

### 2.6. Administration of CGRP-Targeted Therapies

The administration of CGRP-targeted therapies was guided by the expert opinions of pediatric and adolescent headache specialists. Initial doses of CGRP mAbs were administered during inpatient hospitalization to closely monitor for any acute adverse reactions and ensure patient safety during the initiation of treatment. For galcanezumab and fremanezumab, subcutaneous injections were administered at least 5 cm away from the umbilicus to minimize irritation and optimize absorption. Atogepant, an oral CGRP receptor antagonist, was administered in the evening, with dosing schedules tailored to the patient’s clinical profile and needs. Galcanezumab was administered as a loading dose of 240 mg, followed by a monthly dose of 120 mg, while fremanezumab was administered at a monthly dose of 225 mg. Atogepant was taken orally at a dose of 60 mg/day [7,8,9].

The choice of CGRP-targeted therapy was flexible and could be adjusted based on patient tolerance and treatment response. If a specific agent caused intolerable side effects or was deemed ineffective, an alternative was tested to ensure the best possible therapeutic outcome. All patients underwent a mandatory treatment period of 6 months, during which the effects and tolerability were closely monitored. After this period, the continuation or discontinuation of therapy was determined based on the patient’s clinical response and overall condition. Medication changes were guided by clinical response and tolerability; switching occurred after inadequate response over 3 months. This adaptive, patient-centered approach aimed to maximize the benefits of CGRP-targeted therapies while addressing individual needs and safety considerations [13,14].

### 2.7. Lifestyle Modification

Along with the administration of CGRP mAbs, lifestyle modifications were implemented in three key areas—sleep, nutrition, and exercise—which are well-established in the literature as beneficial for managing migraine [21,22,23,24]. These modifications were introduced sequentially, with one intervention added each month, prioritizing tolerability and the potential for migraine improvement in pediatric patients. Interventions were introduced monthly (sleep, nutrition, exercise) to enhance compliance and reduce behavioral burden (Figure 2). The lifestyle index was calculated monthly, with each of the three domains—sleep, nutrition, and exercise—scored from 0 to 3 based on adherence: 0 points for 0–7 days, 1 point for 8–14 days, 2 points for 15–21 days, and 3 points for 22 or more days. The cumulative score across these three domains formed the lifestyle index, ranging from 0 to 9 points, which was assessed at 3-month intervals to track progression and adherence to the prescribed lifestyle modifications [21,22,23,24].

The gradual fading of the blue color in the CGRP-targeted therapy section represents the tapering out of medication in response to symptom improvement over time. The progressively deepening green shades in the sleep, nutritional, and exercise intervention sections indicate the stepwise implementation of lifestyle modifications at monthly intervals.

*Sleep Intervention*: From the start of treatment, patients were guided to improve their sleep quality and consistency. This included exposure to sunlight for at least 30 min in the morning to regulate circadian rhythms, administration of oral melatonin (2–4 mg) to support sleep initiation, and adherence to a bedtime routine with a fixed sleep schedule. Patients were instructed to maintain consistent bedtimes (10:00–11:00 PM) and wake times, ensuring a total sleep duration of 7–9 h per night. Counseling on sleep hygiene practices was provided, including maintaining a dark, quiet bedroom environment and avoiding stimulants close to bedtime [21].

*Nutritional Intervention*: Starting in the second month, patients were encouraged to adopt a low-glycemic index diet to minimize dietary triggers and improve metabolic stability. This involved reducing high-glycemic carbohydrates while ensuring sufficient hydration throughout the day. Educational materials and personalized guidance were provided to families to support adherence to dietary recommendations [22,23].

*Exercise Intervention*: By the third month, patients were advised to incorporate regular physical activity, starting with light aerobic exercises such as daily 30 min walks. These exercises were complemented by gentle stretching exercises targeting the neck and shoulders to relieve tension and promote musculoskeletal health. Physical activity was tailored to each patient’s abilities and gradually adjusted to enhance adherence and effectiveness [24].

### 2.8. Statistical Analysis

All statistical analyses were conducted by two independent medical researchers to ensure objectivity and minimize bias. No formal sample size calculation was conducted due to the exploratory nature of this study. Descriptive statistics were used to summarize the baseline demographic and clinical characteristics, providing an overview of the study population. The Wilcoxon signed-rank test was applied to assess changes over time within the same group, while the Wilcoxon rank-sum test was used to compare differences between specific time intervals for each variable. Time-dependent changes in headache frequency, intensity, and associated disabilities were visually represented using graphs, offering a clear depiction of trends throughout the follow-up period. All analyses were performed using R software version 4.4.2, ensuring robust and reproducible statistical results. Statistical significance was defined as *p* < 0.05.

## 3. Results

### 3.1. Baseline Characteristics of the Study Population

This study included 10 pediatric patients with CM, predominantly female (80%), with a median weight of 55.0 (range: 45.0–71.4) kg. The median age at migraine onset was 10.6 years, and the median age at the first dose of CGRP-targeted therapies was 14 (range: 12–16) years, with a median delay of 18 (range: 3.9–43.0) months from diagnosis to treatment initiation. All brain MRI/MRA and EEG findings were normal, including in the four participants reporting auras. The median baseline PedMIDAS score was 58.5 (range: 32–90), and the migraine frequency was 26.5 (range: 18–28) days per month, reflecting a significant disease burden. Sleep disturbance was reported by nine patients, commonly presenting as difficulty initiating sleep, frequent nighttime awakenings, and irregular sleep-wake schedules; the initial lifestyle index was notably low, with a median of 0.5 (range: 0–1), indicating limited adherence to healthy lifestyle habits. Neuropsychological assessments revealed that patients generally demonstrated adequate cognitive and social functioning, with a median FSIQ of 103 (range: 92–118), MQ of 100 (range: 49–113), and SQ of 103 (range: 98–115). However, depressive symptoms were notable in this cohort, as reflected by a median PHQ-9 score of 18.5 (range: 8–26) (Table 1).

### 3.2. Changes in Migraine-Related Parameters over Time After CGRP-Targeted Therapies

The PedMIDAS score demonstrated a significant reduction over 12 months, decreasing from a median of 58.5 at baseline to 48.0 at 12 months (*p* = 0.037). Migraine frequency also showed a substantial decline, from a median of 26.5 days per month at baseline to 14.0 days at 12 months (*p* < 0.001), accompanied by a reduction in overall response variability. The number of days requiring acute analgesic use decreased significantly from a median of 14 to 5 days (*p* < 0.001), with fewer outliers observed at later time points, reflecting more consistent treatment outcomes. At the initiation of CGRP-targeted therapy, patients continued their existing preventive medications; however, based on clinical improvement during follow-up, several patients were able to taper or discontinue these concomitant therapies under medical supervision.

There was no significant change in GAD-7 scores over 12 months (*p* = 0.788), indicating stability in mild anxiety levels. Conversely, the PHQ-9 score decreased significantly from baseline to 12 months, with a median reduction from 10 to 5 (*p* = 0.045). CDI scores did not change significantly over 12 months, with a median reduction from 20 to 18 (*p* = 0.543), indicating stable depressive symptoms. Similarly, the RCMAS score demonstrated a decreasing trend from a median of 25 to 20 but did not reach statistical significance (*p* = 0.088). The lifestyle index showed marked improvement, increasing significantly from a median of 0 at baseline to 7 at 12 months (*p* < 0.001), reflecting enhanced adherence to the interventions (Figure 3).

### 3.3. Side Effects of CGRP-Targeted Therapies

Six patients required changes in CGRP-targeted therapy during the study period. Of these, three patients switched medications due to reduced efficacy over time, with two transitioning from galcanezumab to fremanezumab and one from fremanezumab to galcanezumab. The remaining three patients switched to atogepant due to injection site pain, with two switching from galcanezumab and one from fremanezumab. Common adverse effects included pain and redness at the injection site (n = 5), myalgia (n = 1), and dizziness (n = 2). No serious adverse events were observed (Table 2).

## 4. Discussion

This study investigated the combined use of CGRP-targeted therapies and lifestyle modifications for managing pediatric chronic migraines, demonstrating their potential effectiveness in this population. Over the course of 12 months, significant improvements were observed in migraine-related disability, as evidenced by a statistically significant reduction in PedMIDAS scores and the frequency of migraine episodes. Additionally, the number of days on which acute analgesics were required decreased markedly, reflecting improved overall migraine management. These findings align with the growing body of evidence supporting the efficacy of CGRP-targeted therapies in reducing the migraine burden, particularly when combined with holistic lifestyle interventions. In particular, this study highlights the tolerability and safety of CGRP-targeted therapies in pediatric patients. While minor side effects such as injection site pain and dizziness were reported, no serious adverse events were observed, reinforcing the potential of these therapies as viable options for children with refractory migraines. Furthermore, the lifestyle index, introduced to quantify adherence to sleep, diet, and exercise interventions, demonstrated progressive improvement over time, highlighting the feasibility of integrating structured lifestyle modifications into pediatric migraine care. These results underscore the importance of a multidisciplinary approach that addresses not only pharmacological needs but also broader determinants of health [1,10,22].

Diagnosing and treating chronic migraine (CM) in pediatric patients is challenging due to nonspecific symptoms, coexisting conditions, and developmental differences that complicate symptom reporting. Applying ICHD-3 criteria is particularly difficult in this age group [11,25]. Therapeutically, the situation is equally challenging due to the limited availability of evidence-based treatment options. Many preventive and acute therapies approved for adults lack robust pediatric-specific trials, leading to off-label use, which raises concerns about efficacy, safety, and tolerability [9,26]. The potential for adverse effects, long-term safety concerns, and medication overuse further restrict the use of available pharmacological treatments, leaving a significant proportion of patients undertreated or untreated [25]. These challenges underscore the critical need for innovative therapeutic approaches, paving the way for the exploration of CGRP-targeted therapies, which show promise in addressing these gaps and are currently undergoing pediatric-focused clinical trials.

CGRP-targeted therapies are emerging as promising treatment options for pediatric and adolescent migraines, addressing a critical unmet need in this population. CGRP, a key neuropeptide implicated in the pathophysiology of migraines, plays a significant role in both adult and younger patients, making it a relevant therapeutic target across different age groups [27,28]. Ongoing clinical trials are evaluating the safety and efficacy of CGRP inhibitors, including monoclonal antibodies and antagonists, in pediatric populations. Preliminary findings, along with off-label studies, suggest that these therapies are both effective and well tolerated in children, with safety profiles comparable to those observed in adults [12,14,29]. Expert opinions further support the potential of CGRP-targeted therapies as valuable options for chronic and refractory migraines in younger patients. In this study, CGRP-targeted therapies demonstrated significant improvements in migraine-related disability, frequency, and analgesic use, while maintaining a favorable safety profile [10]. Further, CGRP-targeted therapies demonstrated both safety and effectiveness in managing chronic migraines in pediatric patients. No serious adverse events were reported, and mild side effects such as injection site pain and dizziness were infrequent and self-limiting, further highlighting the favorable tolerability of these therapies. Significant improvements in migraine-related disability, frequency, and analgesic use observed over the study period align with growing evidence supporting the efficacy of CGRP inhibitors. These findings underscore the potential of CGRP-targeted therapies as safe and effective options for pediatric migraine management, offering meaningful clinical benefits to this vulnerable population [29].

Lifestyle modification is a critical component in managing chronic neurological conditions such as migraines, where sustainable improvements in quality of life require addressing sleep, diet, and exercise [30]. Lifestyle modifications—such as improved sleep hygiene, low-glycemic index diets, and regular aerobic exercise—are increasingly supported by evidence for reducing migraine frequency and severity while enhancing overall well-being [21,24,30,31]. This study demonstrated that CGRP-targeted therapies provided a strong foundation for enabling lifestyle modifications, as headache relief facilitated better adherence to non-pharmacological interventions. Preventive treatments, such as CGRP-targeted therapies, naturally transition patients toward sustained lifestyle changes, reduce dependence on medications, and foster healthier, more sustainable outcomes. The introduction of a lifestyle index allowed for a structured evaluation, showing consistent improvements in lifestyle scores alongside reductions in headache frequency and severity. Notably, two patients discontinued CGRP-targeted therapies after 6 months while maintaining headache control and reducing analgesic use, highlighting the potential for lifestyle changes to sustain therapeutic benefits. These findings suggest that CGRP-targeted therapies may support improved adherence to lifestyle modifications; however, in the absence of direct patient-reported data, this relationship should be interpreted with caution. Nonetheless, integrating lifestyle changes with pharmacological treatments remains a potentially valuable approach in the management of chronic pediatric migraines [30].

This is the first study to investigate CGRP-targeted therapies for chronic migraine in pediatric patients in the Republic of Korea. Although retrospective in nature, this study followed a prospective treatment and observational design, providing valuable insights into the efficacy and safety of CGRP-targeted therapy in this population. A notable strength of this study was the integration of a lifestyle index to evaluate the transition from pharmacological treatment to sustained lifestyle modifications, highlighting the potential of CGRP-targeted therapies to promote long-term health benefits. However, this study has some limitations, including its retrospective design and the off-label use of CGRP-targeted therapies, as these treatments have not yet been FDA-approved for pediatric use. Additionally, this study was an open-label, observational analysis with a small sample size, which may limit the generalizability of the findings and introduces the possibility of bias due to the absence of blinding, particularly in the evaluation of self-reported outcomes. We also acknowledge the potential for placebo effects to have influenced the outcomes, particularly given the pediatric population and reliance on self-reported measures. Future research should address these gaps through larger controlled trials to confirm these findings and explore long-term outcomes. As clinical trials for CGRP-targeted therapies in pediatric populations have concluded, further studies are needed to evaluate their safety and effectiveness in broader age ranges, including those below the approved age. Given the similarities in migraine pathophysiology between adults and children, the introduction of CGRP-targeted therapies holds promise for improving migraine prevention, enabling lifestyle modifications, and fostering positive brain health outcomes in pediatric patients [30,32].

## 5. Conclusions

This study represents the first investigation in Korea to examine the use of CGRP-targeted therapies combined with lifestyle modifications for chronic pediatric migraine. The findings provide preliminary evidence supporting their safety and effectiveness, with observed reductions in migraine-related disability, frequency, and analgesic use. Given the small sample size and retrospective design, larger prospective studies are needed to confirm these results and evaluate long-term outcomes in broader pediatric populations.

## Figures and Tables

**Figure 1 brainsci-15-00493-f001:**
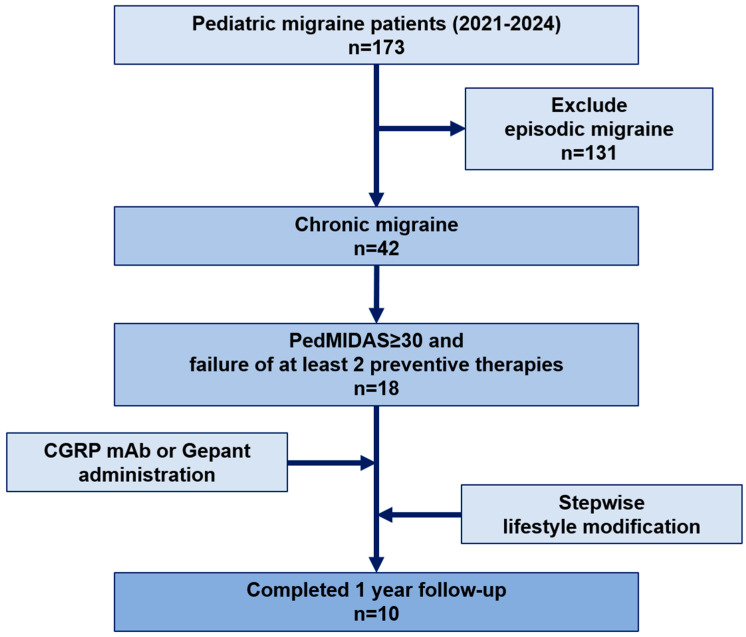
Flowchart of patient selection and study design for chronic pediatric migraine management. Abbreviations: PedMIDAS, Pediatric Migraine Disability Assessment Scale.

**Figure 2 brainsci-15-00493-f002:**
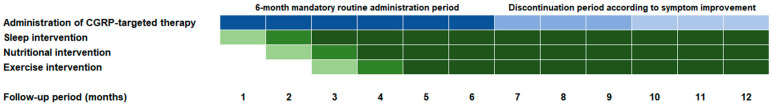
Timeline of CGRP-targeted therapy and stepwise lifestyle modifications over 12-month follow-up period.

**Figure 3 brainsci-15-00493-f003:**
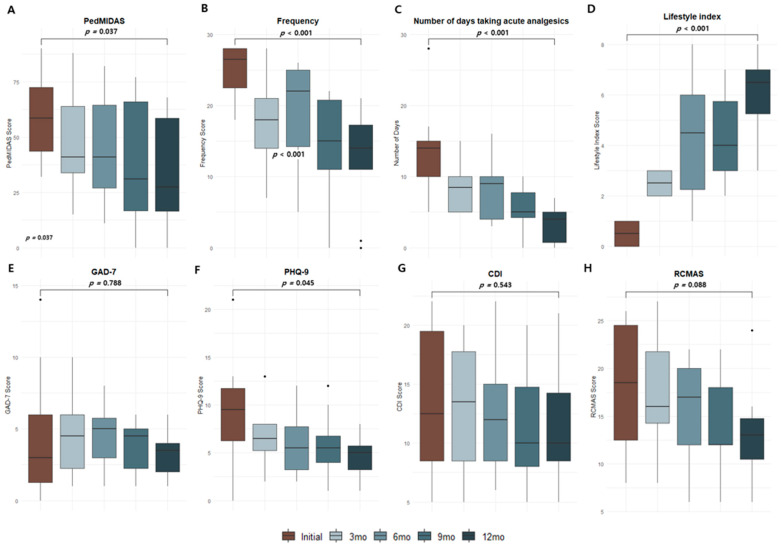
Longitudinal changes in migraine-related outcomes, lifestyle modifications, and psychological assessments over 12 months. This figure illustrates the longitudinal changes in migraine-related disability, lifestyle modifications, and psychological assessments over 12 months. PedMIDAS scores and migraine frequency showed significant reductions, indicating improved migraine-related disability and symptom control. Days requiring acute analgesics also decreased substantially, reflecting better management and reduced medication dependency. The lifestyle index demonstrated consistent improvements, highlighting enhanced adherence to sleep, nutrition, and exercise interventions. Psychological assessments revealed significant improvement in PHQ-9 scores, indicating reduced depressive symptoms, while GAD-7, CDI, and RCMAS scores showed modest or non-significant changes, reflecting stable or slightly improved psychological well-being. The y-axis for frequency, number of days taking acute analgesics, and lifestyle index represents monthly values, whereas the remaining variables reflect measurements taken at 3-month intervals. Abbreviations: PedMIDAS, Pediatric Migraine Disability Assessment Scale; PHQ-9, Patient Health Questionnaire-9; GAD-7, Generalized Anxiety Disorder-7; CDI, Children’s Depression Inventory; RCMAS, Revised Children’s Manifest Anxiety Scale. Figure shows changes in the following parameters over time: (**A**) PedMIDAS score, (**B**) Frequency of migraine episodes, (**C**) Number of days taking acute analgesics, (**D**) Lifestyle index, (**E**) GAD-7 score, (**F**) PHQ-9 score, (**G**) CDI score, and (**H**) RCMAS score.

**Table 1 brainsci-15-00493-t001:** Baseline data of chronic pediatric migraine patients followed for 1 year post initiation of CGRP-targeted therapies.

General Characteristics	N = 10
Sex	M:F = 2:8
Weight (kg)	55.0 (45.0–71.4)
Onset of headache (year)	10.6 (8.1–14.4)
Age at diagnosis of chronic migraine (year)	13.8 (9.6–14.9)
Age at first dose of CGRP-targeted therapies (year)	14 (12–16)
Time from diagnosis of chronic migraine to administration of CGRP-targeted therapies	18.0 (3.9–43.0)
Presence of aura	4
Brain MRI + MRA findings (N = 10)	Normal (N = 10)
EEG findings (n = 4)	Normal (n = 4)
Genetic test findings (n = 4)	Normal (n = 4)
**Baseline characteristics of migraine**	
Average number of analgesics taken (per month)	14 (5–28)
Conventional preventive treatment ever tried	4.5 (3–7)
Pattern of migraine	Pulsating (N = 10)
PedMIDAS at diagnosis (median, range)	58.5 (32–90)
Duration of migraine (median, range)	4 (3–18)
Intensity of migraine (median, range)	7.5 (6–9)
Frequency of migraine (median, range)	26.5 (18–28)
Prior history of head trauma	none
Presence of sleep disturbance	9
Other neurological problems	none
**Initial l** **ifestyle index**	0.5 (0–1)
**Neuropsychological test at diagnosis**	
Full-Scale Intelligence Quotient (FSIQ) (median, range)	103 (92–118)
Memory Quotient (MQ) (median, range)	100 (49–113)
Social Quotient (SQ) (median, range)	103 (98–115)
Patient Health Questionnaire-9 (PHQ-9) (median, range)	18.5 (8–26)
Generalized Anxiety Disorder-7 (GAD-7) (median, range)	3 (0–14)
Children’s Depression Inventory (CDI) (median, range)	12.5 (5–22)
Revised Children’s Manifest Anxiety Scale (RCMAS) (median, range)	18.5 (8–26)
**Initial choice of CGRP-targeted therapies**	
Galcanezumab	6
Fremanezumab	4

**Table 2 brainsci-15-00493-t002:** Changes in CGRP-targeted therapies and their respective side effects.

**Change of Medication**	**Number of** **Patients**	**Reason for Changing Medication**
Galcanezumab -> Fremanezumab	2	Less effective over time
Fremanezumab -> Galcanezumab	1	Less effective over time
Galcanezumab -> Atogepant	2	Pain and redness at injection site, myalgia
Fremanezumab -> Atogepant	1	Pain and redness at injection site
Galcanezumab -> discontinuation (after the 6th dose)	1	Symptom improvement
Fremanezumab -> discontinuation (after the 8th dose)	1	Symptom improvement
**Side Effect**	**Number of** **Patients**	**Types of Side Effects**
Galcanezumab	3	Pain and redness at injection site
	1	myalgia
Fremanezumab	2	Pain and redness at injection site
Atogepant	2	dizziness
Serious adverse event	none	

## Data Availability

Restrictions apply to the datasets. The datasets presented in this article are not readily available because the data are part of an ongoing study. Requests to access the datasets should be directed to [jhnamd83@yuhs.ac].

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
