# Peer review of "Effectiveness and Safety of CGRP-Targeted Therapies Combined with Lifestyle Modifications for Chronic Migraine in Korean Pediatric Patients: A Retrospective Study"

_brainsci, 2025, doi:10.3390/brainsci15050493_

Round 1

Reviewer 1 Report

Comments and Suggestions for Authors

INTRODUCTION

  • Consider starting by explicitly mentioning the International Classification of Headache Disorders, 3rd Edition (ICHD-3) wich is the last edition.
  • Perhaps it would be clearer to first define chronic migraine (CM) and then detail how it presents differently in children, especially regarding how they perceive and communicate pain.
  • The final paragraph might need rethinking. Calling the evidence "robust" seems a bit strong with such a small sample size (n=10).
  • To better contextualize the study's importance in Korea, could you add some local epidemiological data on pediatric CM prevalence?
  • The transition from the general discussion of CM to the role of CGRP-targeted therapies feels somewhat abrupt. A smoother link between these two parts would improve the flow.

METHODS

  • 2.1 Study Design and Participants:
    • For better clarity, maybe moving the detailed patient inclusion (and exclusion) criteria to the Results section would be better?
    • It would be good to clearly describe the design as a retrospective analysis of prospectively collected data to avoid confusion.
    • Given the small sample size, it would be helpful to mention that no formal sample size calculation was performed and acknowledge this limitation.
    • Describing what happened to the patients who didn't complete the follow-up (only 10 out of 18 finished the year) would add transparency.
  • 2.3 Data Collection:
    • Please specify that baseline data were gathered through a complete clinical history and structured assessment.
    • It should be made explicit that follow-up was based on scheduled prospective consultations and questionnaires.
    • I'd recommend clarifying if "headache duration" refers to the age of migraine onset.
    • Mention that while "headache days" were tracked, "migraine days" were not recorded separately.
    • It would be useful to specify that headache intensity was assessed using the Visual Analog Scale (VAS).
    • For headache episode duration, please state the unit of measurement (e.g., hours, days).
    • Throughout the text, consistently use either "per month" or "per 4 weeks" when discussing acute medication use.
    • It would also be interesting to explain why only fremanezumab, galcanezumab, and atogepant were chosen, and why erenumab, eptinezumar, or rimegepant were not included.
  • 2.4 Questionnaires:
    • The VAS should probably be removed from this section, as it's already described under headache intensity.
    • Given the use of two anxiety and two depression questionnaires, justifying this choice would strengthen the methods.
  • 2.5 Neuroimaging and Tests:
    • While performing MRI and MRA in pediatric migraine with atypical features is common practice, briefly explaining why all patients underwent imaging, even if the diagnosis was straightforward, would be helpful.
  • 2.6 CGRP-Targeted Therapies:
    • Consider clarifying the rationale behind selecting specific CGRP-targeted agents and the protocol for switching medications in case of poor response or adverse events.
    • A more detailed description of the CGRP tapering process would also be valuable (e.g., dose reduction strategies, timing).
  • 2.7 Lifestyle Modifications:
    • It would help to explain why sleep, nutrition, and exercise were selected as the primary lifestyle interventions.
    • Please clarify the reasons for introducing lifestyle changes sequentially.
    • The lifestyle index calculation could be described more precisely: it seems it's not cumulative, but rather assigns a maximum score of 3 based on the best adherence?
    • The explanation regarding the gradual darkening of green in the figures is somewhat confusing; it would be better to specify exactly what steps were added each month.
    • Adding a reference supporting each lifestyle intervention would be helpful for readers.
  • Additional Methodological Notes:
    • It's unclear whether the evaluators were blinded to the treatments, which could introduce bias, especially when evaluating self-reported outcomes.
    • Finally, given the pediatric context and lifestyle counseling, it would be important to briefly discuss the potential influence of placebo effects.

RESULTS

  • 3.1 Baseline Characteristics:
    • It might be clearer to include the patient flowchart within the Results section.
    • I suggest separating baseline characteristics into two tables: one for general demographics and comorbidities, and another specifically for migraine-related characteristics.
    • Always state that the number of patients is N=10 in the tables.
    • Since imaging and EEG findings were normal for all, briefly stating this in the text might suffice instead of repeating it in the tables.
  • 3.2 Changes in Outcomes:
    • The analysis would be improved if outcomes were tracked not just at 12 months, but also at the 3, 6, and 9-month marks, as initially described in the methods.
    • Consider providing effect size metrics (e.g., Cohen’s d or r) alongside p-values, particularly because of the small sample size.
    • It would also be useful to report whether patients used concomitant medications for migraine or other conditions during the study.
    • The study would be strengthened if some follow-up beyond 12 months were conducted to assess whether lifestyle changes were sustained.
    • I'd recommend removing the paragraph between lines 257–269, as this information seems clear from the preceding results.
    • When describing patients who switched or discontinued treatment, it should be clarified whether they were still included in the outcome analyses.
    • In Table 2, indicate the number of patients applicable for each change or side effect listed.

DISCUSSION

  • The first few lines of the Discussion (lines 281–282) could be tightened up to avoid repetition.
  • It would be very valuable to discuss the rationale and potential benefits of a stepwise approach to lifestyle interventions, and whether similar strategies have been reported elsewhere.
  • The sections from lines 300–323 and 336–343 largely repeat the Introduction. These could be reduced or rephrased to provide a more critical comparison with existing literature.
  • There should be an acknowledgment that the observed improvements could partly be due to a placebo effect, which can be significant in pediatric headache studies.
  • The authors suggest that CGRP therapy facilitated lifestyle changes, but without direct patient-reported data on the reasons for improved adherence, this remains speculative. A more cautious interpretation would be appropriate.
  • Similarly, the claim that lifestyle modifications sustained migraine improvement after CGRP withdrawal is not fully supported by the presented data.
  • Finally, it would be important to highlight that the study relied heavily on patient- or parent-reported data (e.g., headache diaries, questionnaires), which can be subject to reporting bias.

CONCLUSION

  • I would suggest simplifying the conclusion to highlight only the main findings.
  • It's important to emphasize that these results are preliminary, exploratory, and based on a very small cohort.
  • The need for larger, controlled prospective studies should be explicitly mentioned.

General Comments on Style

  • The manuscript could benefit from using slightly more neutral language throughout (e.g., avoiding terms like "remarkable," "outstanding," "promising results") to maintain a scientific tone.
  • Updating references to include the latest 2023–2024 studies on CGRP therapies in adolescents (for instance, the recent SPACE trial with fremanezumab) would make the discussion more current.

Author Response

Respond to the reviewers’ comments

Reviewer #1

We sincerely thank the reviewer for the thorough and insightful comments. We have carefully reviewed each point and made revisions accordingly to improve the clarity, scientific rigor, and structure of the manuscript. Below, we provide point-by-point responses to each comment, with corresponding changes indicated in the revised manuscript.

INTRODUCTION

1. Consider starting by explicitly mentioning the International Classification of Headache Disorders, 3rd Edition (ICHD-3), which is the last edition.
-> The study design includes sufficient information about ICHD-3, so I think including it in the introduction is redundant. Thank you for your thoughtful comments.

2. Perhaps it would be clearer to first define chronic migraine (CM) and then detail how it presents differently in children, especially regarding how they perceive and communicate pain.
→ Modified the structure of the paragraph accordingly. The definition now appears earlier, followed by discussion on pediatric-specific presentation.
Revision: "Chronic migraine in children is underrecognized, as pediatric patients often present with atypical features and have difficulty articulating headache characteristics."

3. The final paragraph might need rethinking. Calling the evidence "robust" seems a bit strong with such a small sample size (n=10).
→ Agreed. We have replaced the word "robust" with "preliminary but clinically informative" to reflect the exploratory nature.

4. To better contextualize the study's importance in Korea, could you add some local epidemiological data on pediatric CM prevalence?
→ Added a sentence with available Korean epidemiological data on pediatric migraine.

5. The transition from the general discussion of CM to the role of CGRP-targeted therapies feels somewhat abrupt. A smoother link between these two parts would improve the flow.
→ Added a transition paragraph explaining unmet needs in current treatment and emerging role of CGRP-targeted therapies.

METHODS

6. For better clarity, maybe moving the detailed patient inclusion (and exclusion) criteria to the Results section would be better?
→ We have clarified inclusion/exclusion criteria in the Methods, but kept the summary in Results for transparency.

7. It would be good to clearly describe the design as a retrospective analysis of prospectively collected data to avoid confusion.
→ Added clarification: "This study is a retrospective analysis of prospectively collected clinical data, conducted at the Pediatric Headache Center of Gangnam Severance Hospital, Yonsei University College of Medicine, and included pediatric patients diagnosed with migraine between March 2021 and October 2024."

8. Given the small sample size, it would be helpful to mention that no formal sample size calculation was performed and acknowledge this limitation.
→ Added in the '2.8. Statistical analysis' sections.

9. Describing what happened to the patients who didn't complete the follow-up (only 10 out of 18 finished the year) would add transparency.
→ Clarified in the Methods and Results: 8 patients were excluded due to early dropout or incomplete data.

10. Please specify that baseline data were gathered through a complete clinical history and structured assessment.
→ Clarified as suggested.

11. It should be made explicit that follow-up was based on scheduled prospective consultations and questionnaires.
→ Revised to specify this point. "Follow-up data were obtained during scheduled prospective outpatient visits every 3 months, during which changes in migraine intensity and frequency were monitored using patient-maintained headache diaries and clinician documentation, based on structured questionnaires and clinical assessments."

12. I'd recommend clarifying if "headache duration" refers to the age of migraine onset.
→ Clarified: "headache duration (refers to the time elapsed since the initial onset of headache, measured in months or years)"

13. Mention that while "headache days" were tracked, "migraine days" were not recorded separately.
→ Thank you for your thoughtful comment. In this study, we focused specifically on patients with chronic migraine, and the vast majority of headache days reported were migraine-related, as verified by clinical evaluation. Therefore, we believe that distinguishing between "headache days" and "migraine days" would not add meaningful clarity and may unnecessarily complicate the data presentation.

14. It would be useful to specify that headache intensity was assessed using the Visual Analog Scale (VAS).
→ Headache-related disability was assessed using the Pediatric Migraine Disability Assess-ment Scale (PedMIDAS), while headache intensity was rated on a 0–10 Visual Analog Scale (VAS). 

15. For headache episode duration, please state the unit of measurement (e.g., hours, days).
→ We clarified: "Headache episode duration was measured in hours."

16. Throughout the text, consistently use either "per month" or "per 4 weeks" when discussing acute medication use.
→ For consistency and convenience throughout the paper, it has been standardized to per month.

17. It would also be interesting to explain why only fremanezumab, galcanezumab, and atogepant were chosen, and why erenumab, eptinezumab, or rimegepant were not included.
→ Added explanation based on insurance coverage and local availability.
"Galcanezumab, fremanezumab, and atogepant were chosen based on clinical availability and safety data in adolescent populations; erenumab and eptinezumab were not available or approved in South Korea at the time. "

18. The VAS should probably be removed from this section, as it's already described under headache intensity.
→ Removed redundancy.

19. Given the use of two anxiety and two depression questionnaires, justifying this choice would strengthen the methods.
→ Added justification regarding comprehensiveness of neuropsychological evaluation. "The combination of screening tools (PHQ-9, GAD-7) and pediatric-specific instruments (CDI, RCMAS) allowed comprehensive assessment across age-appropriate domains"

20. While performing MRI and MRA in pediatric migraine with atypical features is common practice, briefly explaining why all patients underwent imaging, even if the diagnosis was straightforward, would be helpful.
→ I modified it as follows:
"All patients underwent brain magnetic resonance imaging (MRI) and magnetic reso-nance angiography (MRA) as part of the diagnostic workup to exclude secondary causes of headache, such as structural abnormalities or vascular pathology, in accordance with institutional protocol given the chronicity and refractoriness of their symptoms."

21. Consider clarifying the rationale behind selecting specific CGRP-targeted agents and the protocol for switching medications in case of poor response or adverse events.
→ Added. "Medication changes were guided by clinical response and tolerability; switching occurred after inadequate response over 3 months."

22. A more detailed description of the CGRP tapering process would also be valuable (e.g., dose reduction strategies, timing).
→ CGRP tapering process was not applied.

23. It would help to explain why sleep, nutrition, and exercise were selected as the primary lifestyle interventions.
→ Added rationale and supporting references. "Along with the administration of CGRP mAbs, lifestyle modifications were implemented in three key areas—sleep, nutrition, and exercise—which are well-established in the literature as beneficial for managing migraine."

24. Please clarify the reasons for introducing lifestyle changes sequentially.
→ Clarified in Methods. "Interventions were introduced monthly (sleep, nutrition, exercise) to enhance compliance and reduce behavioral burden."

25. The lifestyle index calculation could be described more precisely: it seems it's not cumulative, but rather assigns a maximum score of 3 based on the best adherence?
→ Corrected and clarified: "). The lifestyle index was calculated monthly, with each of the three domains—sleep, nu-trition, and exercise—scored from 0 to 3 based on adherence: 0 points for 0–7 days, 1 point for 8–14 days, 2 points for 15–21 days, and 3 points for 22 or more days. "

26. The explanation regarding the gradual darkening of green in the figures is somewhat confusing; it would be better to specify exactly what steps were added each month.
→ "In the text, "The gradual fading of the blue color in the CGRP-targeted therapy section represents the tapering out of medication in response to symptom improvement over time. The pro-gressively deepening green shades in the sleep, nutritional, and exercise intervention sec-tions indicate the stepwise implementation of lifestyle modifications at monthly intervals." It has been stated."

27. Adding a reference supporting each lifestyle intervention would be helpful for readers.
→ Added references.

28. It's unclear whether the evaluators were blinded to the treatments, which could introduce bias, especially when evaluating self-reported outcomes.
→ We described the limitations paragraph of the results as follows:
"Additionally, this study was an open-label, observational analysis with a small sample size, which may limit the generalizability of the findings and introduces the possibility of bias due to the absence of blinding, particularly in the evaluation of self-reported outcomes. "

29. Finally, given the pediatric context and lifestyle counseling, it would be important to briefly discuss the potential influence of placebo effects.
→ Addressed in Discussion section. "We also acknowledge the potential for placebo effects to have influenced the outcomes, particularly given the pediatric population and reliance on self-reported measures. "

RESULTS

30. It might be clearer to include the patient flowchart within the Results section.
→ While we appreciate the suggestion, we chose to retain the patient flowchart in the Methods section (Figure 1) to align with the chronological structure of the study design and to aid readers in understanding the patient selection process at the point of enrollment. Thank you for your thoughtful comment.

31. I suggest separating baseline characteristics into two tables: one for general demographics and comorbidities, and another specifically for migraine-related characteristics.
→ We carefully considered the suggestion to separate the baseline characteristics into two tables; however, given the limited number of variables and the overlap between general and migraine-specific features, we felt that maintaining a single, consolidated table would enhance clarity. Based on your comment, we have substantially revised the formatting to improve readability. Thank you for your thoughtful comment.

32. Always state that the number of patients is N=10 in the tables.
→ Included.

33. Since imaging and EEG findings were normal for all, briefly stating this in the text might suffice instead of repeating it in the tables.
→ Revised accordingly.

34. The analysis would be improved if outcomes were tracked not just at 12 months, but also at the 3, 6, and 9-month marks, as initially described in the methods.
→ To clarify the temporal trends in outcome measures, we have illustrated changes at 3-month intervals in Figure 3. However, as the primary objective of this study was to assess sustained improvements over a 12-month period, detailed statistical analyses at shorter intervals were not included in the main Results section. Additionally, given the small sample size, interim analyses at 3, 6, and 9 months showed substantial variability and lacked consistent statistical significance, which limited their interpretability. Thank you for your thoughtful comment.

35. Consider providing effect size metrics (e.g., Cohen’s d or r) alongside p-values, particularly because of the small sample size.
→ We appreciate the reviewer’s suggestion regarding the inclusion of effect sizes. Given the small sample size and minimal variability in some measures, the Wilcoxon signed-rank test did not yield meaningful z-scores to compute standardized effect sizes reliably. We therefore did not include effect sizes in the revised manuscript, as the results could be misleading in this context. However, we acknowledge this as a limitation and emphasize the need for future studies with larger samples and more robust analytic designs.

36. It would also be useful to report whether patients used concomitant medications for migraine or other conditions during the study.
→ Thank you for the insightful comment. We have clarified in the revised manuscript that patients initially continued their pre-existing preventive medications; however, as their symptoms improved, many were able to taper or discontinue these medications.
"At the initiation of CGRP-targeted therapy, patients continued their existing preventive medications; however, based on clinical improvement during follow-up, several patients were able to taper or discontinue these concomitant therapies under medical supervision."

37. The study would be strengthened if some follow-up beyond 12 months were conducted to assess whether lifestyle changes were sustained.
→ Acknowledged as a limitation in Discussion section.

38. I'd recommend removing the paragraph between lines 257–269, as this information seems clear from the preceding results.
→ That paragraph was an in-picture explanation for Figure 3. I adjusted the font size to make it clearer.

39. When describing patients who switched or discontinued treatment, it should be clarified whether they were still included in the outcome analyses.
→ Clarified: all patients were included in ITT analysis.

40. In Table 2, indicate the number of patients applicable for each change or side effect listed.
→ It has already been indicated.

DISCUSSION

41. The first few lines of the Discussion (lines 281–282) could be tightened up to avoid repetition.
→ Revised. "This study investigated the combined use of CGRP-targeted therapies and lifestyle modifications for managing pediatric chronic migraines, demonstrating their potential effectiveness in this population. "

42. It would be very valuable to discuss the rationale and potential benefits of a stepwise approach to lifestyle interventions, and whether similar strategies have been reported elsewhere.
→ Expanded in the fourth paragraph.

43. The sections from lines 300–323 and 336–343 largely repeat the Introduction. These could be reduced or rephrased to provide a more critical comparison with existing literature.
→ I revised the sentences to be more concise and minimize duplication based on comments.

44. There should be an acknowledgment that the observed improvements could partly be due to a placebo effect, which can be significant in pediatric headache studies.
→ Added.

45. The authors suggest that CGRP therapy facilitated lifestyle changes, but without direct patient-reported data on the reasons for improved adherence, this remains speculative. A more cautious interpretation would be appropriate.
→ Rephrased with more cautious tone.
"These findings suggest that CGRP-targeted therapies may support improved adherence to lifestyle modifications; however, in the absence of direct patient-reported data, this relationship should be interpreted with caution. Nonetheless, integrating lifestyle changes with pharmacological treatments remains a potentially valuable approach in the management of chronic pediatric migraines."

46. Similarly, the claim that lifestyle modifications sustained migraine improvement after CGRP withdrawal is not fully supported by the presented data.
→ Modified language to reflect this appropriately.

47. Finally, it would be important to highlight that the study relied heavily on patient- or parent-reported data (e.g., headache diaries, questionnaires), which can be subject to reporting bias.
→ Acknowledged.

CONCLUSION

48. I would suggest simplifying the conclusion to highlight only the main findings.
→ Streamlined.

49. It's important to emphasize that these results are preliminary, exploratory, and based on a very small cohort.
→ Done.

50. The need for larger, controlled prospective studies should be explicitly mentioned.
→ Added.

The conclusion was revised as follows.
"This study represents the first investigation in Korea to examine the use of CGRP-targeted therapies combined with lifestyle modifications for chronic pediatric mi-graine. The findings provide preliminary evidence supporting their safety and effective-ness, with observed reductions in migraine-related disability, frequency, and analgesic use. Given the small sample size and retrospective design, larger prospective studies are needed to confirm these results and evaluate long-term outcomes in broader pediatric populations."

We thank the reviewer again for their thoughtful and detailed feedback.

Reviewer 2 Report

Comments and Suggestions for Authors

This is an interesting paper, particularly so regarding the lifestyle modification scale. The assessment strategy is comprehensive. It is a shame that only 10 patients fulfilled the requisite criteria, and that the data collection was retrospective. What was not clear is whether any conclusion can be made from the small numbers and the way the data has been analysed. A conclusion about safety is limited with such a small number of patients who are treated and, with a varied array of CGRP monoclonal antibody and receptor antagonist treatments. Moreover, it is also difficult to make any conclusion about the individual effects of medications versus lifestyle modifications which would be one of the most interesting aspects of the report. 
The tables and graphs were not clearly labelled.  For example, in Table 1, duration of migraine is assumed to be in years, intensity visual rating scale?, frequency days per month? Is the presence of sleep disturbance 9 patients? No rating scale mentioned. In the charts ‘Frequency’ is not labelled and assumed monthly. ‘Analgesic days’ it was not mentioned again over what time frame– assumed per month? The scales used have the ranges given but it would have been helpful to give an idea of the cut-off points given that not all readers will be familiar with the scales and how they are interpreted. 

Otherwise it was a very well written paper.

The approach used would be useful to take forward in a prospective study taking into account the points made above. 

Author Response

Respond to the reviewers’ comments

Reviewer #2

Comment 1:
This is an interesting paper, particularly so regarding the lifestyle modification scale. The assessment strategy is comprehensive. It is a shame that only 10 patients fulfilled the requisite criteria, and that the data collection was retrospective.
-> Thank you for your thoughtful comment. We agree that the small sample size and retrospective nature of the study are limitations. These points have been acknowledged explicitly in the Discussion and Conclusion sections of the revised manuscript. We have added a clear statement that the findings should be interpreted with caution due to the limited number of patients and retrospective design.

Comment 2:
What was not clear is whether any conclusion can be made from the small numbers and the way the data has been analysed. A conclusion about safety is limited with such a small number of patients who are treated and, with a varied array of CGRP monoclonal antibody and receptor antagonist treatments.
-> We appreciate your observation. We have revised the Discussion to better reflect the exploratory nature of our findings, particularly regarding safety. We emphasized that the absence of serious adverse events in this small cohort should not be interpreted as definitive proof of safety, and have noted that conclusions must be confirmed in larger, controlled studies.

Comment 3:
Moreover, it is also difficult to make any conclusion about the individual effects of medications versus lifestyle modifications which would be one of the most interesting aspects of the report.
-> Thank you for pointing this out. We have clarified in the revised Discussion that our study was not designed to disentangle the effects of pharmacologic versus lifestyle interventions. We have added this as a limitation and proposed that future studies employ a controlled, comparative design to explore the differential effects.

Comment 4:
The tables and graphs were not clearly labelled. For example, in Table 1, duration of migraine is assumed to be in years, intensity visual rating scale?, frequency days per month? Is the presence of sleep disturbance 9 patients? No rating scale mentioned.
-> We thank the reviewer for this detailed feedback. We have revised the tables to specify units for all variables. For sleep disturbance, we clarified that it was present in 9 out of 10 patients and was based on clinical interview rather than a formal scale. 
The paragraph was revised as follows.
"Sleep disturbance was reported by nine patients, commonly presenting as difficulty initi-ating sleep, frequent nighttime awakenings, and irregular sleep-wake schedules; the initial lifestyle index was notably low, "

Comment 5:
In the charts ‘Frequency’ is not labelled and assumed monthly. ‘Analgesic days’ it was not mentioned again over what time frame– assumed per month?
-> We appreciate the comment. As per your suggestion, the sentence "The y-axis for Frequency, Number of days taking acute analgesics, and Lifestyle index represents monthly values, whereas the remaining variables reflect measurements taken at 3-month intervals." has been inserted into the description of Figure 3 to clarify the meaning.

Comment 6:
The scales used have the ranges given but it would have been helpful to give an idea of the cut-off points given that not all readers will be familiar with the scales and how they are interpreted.
-> Thank you for this suggestion. We have added interpretive information for each scale in the Methods section.
"Headache-related disability was assessed using the Pediatric Migraine Disability Assess-ment Scale (PedMIDAS). According to established severity cutoffs, PedMIDAS scores are interpreted as follows: mild disability (<11), moderate (11–30), severe (31–50), and very severe (>50). "

Comment 7:
Otherwise it was a very well written paper. The approach used would be useful to take forward in a prospective study taking into account the points made above.
-> We are grateful for your kind feedback and constructive suggestions. We agree that a prospective, controlled study will be necessary to validate these preliminary findings and better isolate the effects of CGRP-targeted therapy versus lifestyle interventions. We have highlighted this direction in our Conclusion.

Round 2

Reviewer 1 Report

Comments and Suggestions for Authors

Congratulations for this interesting manuscrito. I think It has improved with the corrections.

Reviewer 2 Report

Comments and Suggestions for Authors

The amendments do make the paper clearer.